# GAPX: Generalized Autoregressive Paraphrase-Identification X

**Yifei Zhou**
Cornell University
yz639@cornell.edu

**Renyu Li**
Cornell University
rl626@cornell.edu

**Hayden Housen**
Cornell University
hth33@cornell.edu

**Ser-nam Lim**
Meta AI
sernamlim@fb.com

## Abstract

Paraphrase Identification is a fundamental task in Natural Language Processing. While much progress has been made in the field, the performance of many state-of-the-art models often suffer from distribution shift during inference time. We verify that a major source of this performance drop comes from biases introduced by negative examples. To overcome these biases, we propose in this paper to train two separate models, one that only utilizes the positive pairs and the other the negative pairs. This enables us the option of deciding how much to utilize the negative model, for which we introduce a perplexity based out-of-distribution metric that we show can effectively and automatically determine how much weight it should be given during inference. We support our findings with strong empirical results. [1]

## 1 Introduction

Paraphrases are sentences or phrases that convey the same meaning using different wording, and is fundamental to the understanding of languages [7]. Paraphrase Identification is a well-studied task of identifying if a given pair of sentences has the same meaning [51, 47, 56, 57, 31], and has many important downstream applications such as machine translation [61, 44, 40, 27], and question-answering [11, 35].

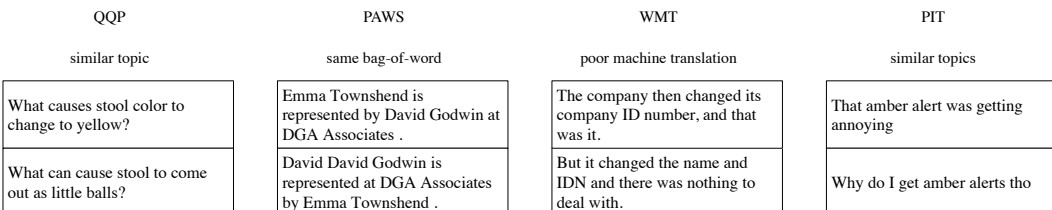

Figure 1: Negative pairs in different datasets are mined differently in different datasets, and can lead to significant biases during training.

Recently, researchers have observed that neural network architectures trained on different datasets could achieve state-of-the-art performances for the task of paraphrase identification [52, 16, 50]. While these advances are encouraging for the research community, it has however been observed

---

[1]Our code is publicly available at: https://github.com/YifeiZhou02/generalized_paraphrase_identification

36th Conference on Neural Information Processing Systems (NeurIPS 2022).

that these models can be especially fragile in the face of distribution shift [45]. In other words, when a model trained on a source dataset $\mathcal{D}^s$ is tested on another dataset, $\mathcal{D}^t$, collected and annotated independently, and with a distribution shift, the classification accuracy drops significantly [62].

This paper presents our findings and observations that negative pairs (i.e., non-paraphrase pairs) in the training set, as opposed to the positive pairs, do not generalize well to out-of-distribution test pairs. Intuitively, negative pairs only represent a limited perspective of how the meanings of sentences can be different (and indeed it is practically infeasible to represent every possible perspective). We conjecture that negative pairs are so specific to the dataset that they adversely encourage the model to learn biased representations. We show this observation in Figure 1. Quora Question Pair (QQP) [2] extracts its negative pairs from similar topics. Paraphrase Adversarials from Word Scrambling (PAWS) Zhang et al. [62] generate negative pairs primarily from word swapping. World Machine Translation Metrics Task 2017 (WMT) [8] considers negative examples as poor machine translations. We therefore hypothesize that biases introduced by the different ways negative pairs are mined are major causes of the poor generalizability of paraphrase identification models.

Based on this observation, we would like to be able to control the reliance on negative pairs for out-of-distribution prediction. In order to achieve this, we propose to explicitly train two separate models for the positive and negative pairs (we will refer to them as the positive and negative models respectively), which will give us the option to choose when to use the negative model. It is well known that just training on positive pairs alone can lead to a degenerate solution [21, 59, 42, 13], e.g., a constant function network would still produce a perfect training loss. To prevent this, we propose a novel generative framework where we use an autoregressive transformer [39, 49], specifically BART [30]. Given two sentences, we condition the prediction of the next token in the second sentence on the first sentence and the previous tokens. In a Bayesian sense, this would mean that the next token predicted has a higher probability of being a positive/negative pair to the first sentence for the positive and negative model respectively. This learning strategy has no degenerate solutions even when we are training the positive and negative models separately. We call our proposed approach GAP to stand for Generalized Autoregressive Paraphrase-Identification. One potential pitfall of GAP is that it ignores the "interplay" between positive and negative pairs that would otherwise be learned if they are utilized in training together. This is especially important when the test pairs are in-distribution. To overcome, we utilize an extra discriminative model, trained with both positive and negative pairs, to capture the interplay. We call this extension GAPX (pronounced as "Gaps") to capture the eXtra discriminative model used.

For all practical purposes, the weights we placed on the positive, negative and/or discriminative model in GAP and GAPX need to be determined automatically during inference. For in-distribution pairs, we desire to use them all, while for out-of-distribution pairs, we hope to rely on the positive model much more heavily. This obviously leads to a question of how to determine whether a given pair is in or out of distribution [9, 14, 43, 17, 20]. During testing, our method ensembles the positive model, the negative model, and the discriminative model based on the degree of the similarity of the test pair to the training pairs, and found that this works well for our purpose. We measure this degree of similarity with probability cumulative density function (cdf) in terms of perplexity [25], and show that it is superior to other measures.

In summary, our contributions are as follow:

1. We report new research insights, supported by empirical results, that the negative pairs of a dataset could potentially introduce biases that will prevent a paraphrase identification model from generalizing to out-of-distribution pairs.

2. To overcome, we propose a novel autoregressive modeling approach to train both a positive and a negative model, and ensemble them automatically during inference. Further, we observe that the interplay between positive and negative pairs are important for in-distribution inference, for which we add a discriminative model. We then introduce a new perplexity based approach to determine whether a given pair is out-of-distribution to achieve auto ensembling.

3. We support our proposal with strong empirical results. Compared to state-of-the-art trans-formers in out-of-distribution performance, our model achieves an average of 11.5% and 8.4% improvement in terms of macro F1 and accuracy respectively over 7 different out-

---

[2]https://quoradata.quora.com/First-Quora-Dataset-Release-Question-Pairs

of-distribution scenarios. Our method is especially robust to paraphrase adversarials like PAWS, while keeping comparable performance for in-distribution prediction.

## 2 Related Works

### 2.1 Distribution Shift and Debiasing Models in NLP

The issue of dataset bias and model debiasing has been widely studied in a lot of field in NLP such as Natural Language Inference [23, 3] and Question Answering [36, 2, 4]. Notable work by [22, 5, 10, 15, 48] utilize ensembling to reduce models' reliance on dataset bias. These models share the same paradigm where they break down a given sample $x$ into signal feature $x_s$ and biased feature $x_b$, in the hope of preventing their model from relying on $x_b$, which has been shown to be the limiting factor preventing the model from generalizing to out-of-distribution samples [5]. Here, a separate model is first either trained on $x_b$ or on datasets with known biases [10, 15, 22], or acquired from models known to have limited generalization capability. Then they train their main model with a regularization term that encourages the main model to produce predictions that deviate from that of the "biased model". However, this type of approach has shown limited success [45] in debiasing paraphrase identification models. In contrast, our method is based on our observation that negative pairs limit the generalization of paraphrase identification models.

### 2.2 Out-of-distribution Detection

Another line of work relevant to this paper is the task of detecting out-of-distribution samples [9, 14, 43, 17, 20, 55]. Researchers have proposed methods to detect anomaly samples by examining the softmax scores [32] or energy scores [33, 60] produced by discriminative models, while others take a more probabilistic approach to estimate the probability density [14, 25, 66, 1] or reconstruction error [38]. In this paper, we introduce a novel perplexity based out-of-distribution detection method that we show empirically to work well for our purpose. Specifically, during inference, an out-of-distribution score is utilized to weigh the contributions from the positive and negative models: the higher the confidence that the sample is out-of-distribution, the lesser the negative model's contribution.

### 2.3 Text Generation Metrics

Finally, we would like to note the difference between our work and autoregressive methods that have been explored [58, 46] for evaluating text generation. Our work differs as follows: 1) Paraphrase identification seeks to assign a label of paraphrase or not while text generation metrics seeks to assign a score to measure the similarity of sentences; 2) Current text generation metrics either cannot be trained to fit to a specifc distribution [61, 58, 63] or are limited to the i.i.d. setting [44, 41] of the training distribution. In contrast, our method not only significantly improves out-of-distribution performances but is also competitive with state-of-the-art paraphrase identification methods for in-distribution predictions.

## 3 Methodology

We observe that negative pairs in paraphrase identification constitute the main source of bias. To overcome this, we propose the following training paradigms to learn a significantly less biased paraphrase identification model. We employ an autoregressive conditional sentence generators with transformer architecture as the backbone of our model. Specifically, we train a positive and negative model to estimate the distribution of positive and negative pairs in a dataset respectively. During testing, the two models are ensembled based on how likely the input pair is out of distribution. This section provides details on our method.

### 3.1 Separation of Dependence on Positive and Negative Pairs

Let $\mathcal{S}$ be the space of all sentences, $X = (s_1, s_2)$ be the random variable representing a sample pair from $\mathcal{S}$, and $Y$ the random variable representing the labels, with $Y = 1$ indicating that $s_1$ and $s_2$ are paraphrases and otherwise when $Y = 0$. We seek to separate the dependence between the distribution of positive and negative pairs, motivated by the observation of the presence of bias in the negative

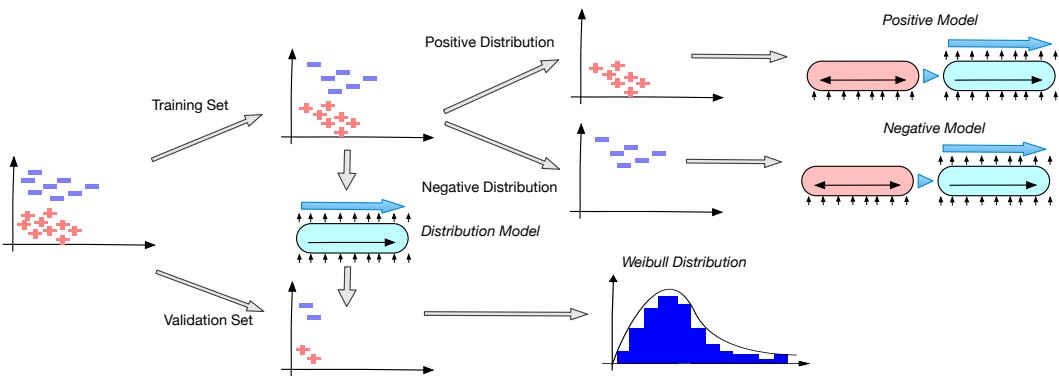

Figure 2: An overview of the training procedure of our model GAP. GAPX ensembles GAP with another discriminative model.

pairs. To begin, we model the distribution of sentences by splitting the sentence $s_2$ of length $n$ into the autoregressive product of individual words, where $w_2^{(i)}$ denotes the $i$th word in $s_2$. By applying Bayesian Inference Law, we have:

$$P(Y = y|s_1, s_2) = \frac{P(Y = y|s_1)\Pi_{i=1}^n P(w_2^{(i)}|s_1, Y = y, w_2^{(1:i-1)})}{\Pi_{i=1}^n P(w_2^{(i)}|w_2^{(1:i-1)}, s_1)}. \tag{1}$$

Subtracting the logarithm for $Y = 1$ and $Y = 0$, we get:

$$\begin{aligned}
&\log(P(Y = 1|s_1, s_2)) - \log(P(Y = 0|s_1, s_2)) \\
&= (\log P(Y = 1|s_1) - \log P(Y = 0|s_1)) \\
&+ (\sum_{i=1}^n \log P(w_2^{(i)}|s_1, Y = 1, w_2^{(1:i-1)}) - (\sum_{i=1}^n \log P(w_2^{(i)}|s_1, Y = 0, w_2^{(1:i-1)})) \\
&= (\log P(Y = 1) - \log P(Y = 0)) \\
&+ (\sum_{i=1}^n \log P(w_2^{(i)}|s_1, Y = 1, w_2^{(1:i-1)}) - (\sum_{i=1}^n \log P(w_2^{(i)}|s_1, Y = 0, w_2^{(1:i-1)})). \tag{2}
\end{aligned}$$

In this way, we break the probability inference into 3 terms resulting in Eqn. 2: (1) $(\log P(Y = 1) - \log P(Y = 0))$, which should just be a constant; (2) $(\sum_{i=1}^n \log P(w_2^{(i)}|s_1, Y = 1, w_2^{(1:i-1)}))$, which depends only on the distribution of positive pairs; (3)$(\sum_{i=1}^n \log P(w_2^{(i)}|s_1, Y = 0, w_2^{(1:i-1)}))$, which depends only on the distribution of negative pairs. We define the score of confidence as follows:

$$S(s_1, s_2) = \underbrace{(\sum_{i=1}^n \log P(w_2^{(i)}|s_1, Y = 1, w_2^{(1:i-1)})}_{\text{Positive Model}} - \underbrace{(\sum_{i=1}^n \log P(w_2^{(i)}|s_1, Y = 0, w_2^{(1:i-1)}))}_{\text{Negative Model}}. \tag{3}$$

In the above, we are now left with two terms, the first representing the positive model and the second the negative model. If we were to train the two terms together, the effects of the negative pairs in the resulting model can never be removed during inference, which we have observed to be a major source of bias. To avoid this, we propose to train the first term and the second term separately, and then subsequently ensemble them based on the degree that a given pair is out of distribution. We train each model on top of the pretrained autoregressive transformer described in [30] known as BART. Given $s_1$ and $s_2$, we feed $s_1$ into the encoder as the condition, shift $s_2$ to the right by one-token, and feed shifted $s_2$ to the decoder. While the decoder proceeds autoregressively, we record the next-word probability distribution. We calculate the cross entropy between the next-word probability distribution and the target token in $s_2$ to update the model parameters. Note that here the Bayesian formulation has been similarly raised in some of the previous work like Moore and Lewis [37], but to the best of our knowledge, we're the first to propose this Bayesian formulation to control the reliance on different components of the model.

## 3.2 Ensembling

To combine the positive and negative model, if we know a priori $\mathcal{D}^t$ is in the same distribution as $\mathcal{D}^s$, we can directly substitute the prediction of the positive and negative model into Eqn. 3. We will refer to this as the In-distribution Predictor (IDP). If we have reason to believe that there is a significant distribution shift between $\mathcal{D}^s$ and $\mathcal{D}^t$ (e.g., different sources of corpus and different dataset collection procedure), we observe empirically that we should only utilize the positive model and disregard the negative model due to the bias it introduces. We will refer to this as the Out-of-distribution Predictor (OODP).

### 3.2.1 Automatic Ensembling

However, in most cases, we have little or no knowledge of the testing distribution, in which case we need to automatically decide how important the negative model is by detecting how much a test pair is in the same distribution as the training set. We adopt a weighted interpolation between a constant and the negative model in addition to the positive model as follow:

$$S(s_1, s_2) = \log P(s_2|s_1, Y = 1) - (1 - \lambda(s_1, s_2)) \log P(s_2|s_1, Y = 0) - \lambda(s_1, s_2)C, \quad (4)$$

where $\lambda(s_1, s_2)$ is a weight parameter depending on $s_1$ and $s_2$, and $C$ is a constant that achieves a regularization effect. See Appendix for ablations on how $C$ can be set. $P(s_2|s_1, Y = 1)$ and $P(s_2|s_1, Y = 0)$ are the same terms in Eqn.3. To automatically assign $\lambda(s_1, s_2)$ for different sentence pairs, we measure an out-of-distribution score for $(s_1, s_2)$ with regard to the training distribution. Specifically, we use the same set of training data, comprising both positive and negative pairs, from $\mathcal{D}^s$, on which we train another autoregressive model, which we will refer to as the distribution model. The distribution model is trained by feeding an empty string into the encoder and the concatenated $s_1$ and $s_2$ into the decoder, with the training goal of predicting the next token. We measure the perplexity of each sentence pair $(s_1, s_2)$ using the distribution model based on the following formula, $w^i$ being the $i$th token of the concatenated $(s_1, s_2)$ of length $n$:

$$PP(s_1, s_2) = \sqrt[n]{(\prod_{i=1}^{n} \frac{1}{P(w^i|w^{1:i-1})}}. \quad (5)$$

We then fit a Weibull distribution to the perplexity of a held-back set of validation data, so that it can better model the right-skewed property of the distribution. We derive the exponential parameter $a$, the shape parameter $c$, the location parameter $loc$, and the scale parameter $scale$. During testing, $\lambda(s_1, s_2)$ can now be determined as:

$$\lambda(s_1, s_2) = cdf(PP(s_1, s_2), Weibull(a, c, loc, scale)). \quad (6)$$

For the final prediction, we predict the sentence pair to be paraphrase if $S(s_1, s_2) \geq 0$ and non-paraphrase otherwise. This forms what we referred to earlier as GAP (Generalized Autoregressive Paraphrase-Identification).

### 3.2.2 Capturing Interplay Between Positive and Negative Pairs

In practice, training a positive and negative model separately disregards the interplay between the positive and negative pairs, which could be important when the test pairs are in-distribution. To capture such interplay, we utilize both positive and negative pairs to train a discriminative model for sequence classification. Specifically, we first define a thresholding function based on the value of $\lambda$:

$$\tau(\lambda) = \begin{cases} 0 & \lambda < 0.9 \\ 1 & else. \end{cases} \quad (7)$$

We then ensemble the discriminative model and GAP using the value of $\tau(\lambda)$:

$$S^*(s_1, s_2) = M(1 - \tau(\lambda(s_1, s_2)))(P(Y = 1|s_1, s_2) - \frac{1}{2}) + \tau(\lambda(s_1, s_2))S(s_1, s_2), \quad (8)$$

where $P(Y = 1|s_1, s_2)$ can be estimated by any discriminative model, and $M$ is a sufficiently large constant. Note that this definition is essentially the same as trusting the discriminative model when we do not have statistical evidence that the pair is out-of-distribution (p-test < 10%) while trusting the GAP model otherwise. For the final prediction, we predict the sentence pair to be paraphrase if $S * (s_1, s_2) \geq 0$ and non-paraphrase otherwise. This defines GAPX (Generalized Autoregressive Paraphrase Identfication X), for which we set $M$ to be sufficiently large ($> 1000$), so that when comparing the model confidence for different pairs, the score of the discriminative model will be prioritized.

## 4 Experiments

Our experiments are designed to (1) verify that the task of paraphrase identification suffers from biases in the datasets that is the main obstacle to generalization in this field of study, (2) test the accuracy of our perplexity based out-of-distribution detection method, and (3) test that balancing the utilization of the negative model can help outperform the state-of-the-art in the face of distribution shift, without losing in the in-distribution scenarios.

### 4.1 Datasets

We compare our method against the other state-of-the-art methods on different combinations of the following datasets:

- Quora Question Pair (QQP) consists of over 400,000 lines of potential question duplicate pairs. Since the original sampling method returns an imbalanced dataset, the authors attempt to balance it with additional negative pairs collected from similar topics to make them harder. Note that to scale QQP down to approximately the same size of PAWS and PIT (see below), we take the first 10k training pairs and 2k testing pairs from the train and test split by Wang et al. [52].

- World Machine Translation Metrics Task 2017 (WMT) [8] contains in total 3793 manual ratings of machine translations from 7 languages to English. Each rating result contains a source sentence in the source language, a reference sentence in English (ground truth translation), a machine translated sentence in English, and a manual rating of the quality of translations. We take the ground-truth reference sentence and the machine translated sentence as the sentence pair. Sentences with a higher quality score ($> 0$) are labeled as paraphrases, while those with lower quality scores ($\leq 0$) are labeled otherwise. The resulting paraphase identification dataset is balanced. Note that this dataset is significantly smaller than other datasets, so we only use it as a test set.

- Paraphrase and Semantic Similarity in Twitter (PIT) [53, 54] contains 18762 sentence pairs automatically extracted from a similar distribution of topics as QQP. Annotators manually assigned integer scores from 0 to 5 to each sentence pair, representing the degree of similarity between the sentence pair. To make it a paraphrase identification dataset, we label sentence pairs with low scores (0, 1) as non-paraphrases and those with high scores (4,5) as paraphrases. The original dataset is unbalanced, so we randomly sample a maximum balanced subset of the dataset. The original test set processed in this way shrinks to only 350, and is not comparable to the other datasets. Hence, we use the original development data of size 1896 as the test set while keeping original test set of size 350 for development. The training set contains 5332 sentence pairs.

- Paraphrase Adversarials from Word Scrambling (PAWS) [62] contains 49,401 sentence pairs, each of which is constructed from the same bag-of-word to make the evaluation more challenging. Most of the negative pairs are generated by word swapping while positive pairs are supplemented by back translation. This dataset contains paraphrase pairs that are the adversarial counterparts of standard paraphrase datasets such as QQP and PIT.

With these datasets, we perform experiments where different models (Sec. 4.2) are trained on one dataset and evaluated on another in order to observe whether their performance hold in the face of distribution shift.

## 4.2 Benchmarks

We benchmark the paraphrase identification task with these methods:

- **BOW** [62] represents two input sentences with bag of words. The bag of words representation of each input sentence is passed through a fully-connected network and cosine similarity between of the final layer is used to compute the classification output.

- **BiLSTM** [28] passes each of the two input sentences through a bidirectional LSTM network. The output state of the two sentences are then concatenated together and passed through a fully-connected network to get the classification output.

- **BERT** [16] is representative of the state-of-the-art transformer methods for text classification. We finetune the pretrained model "bert-base-uncased" in a standard way. We concatenate the sentence pair separated by a [SEP] token and take the [CLS] token as aggregate representation for the sentence pair. The embedding for the [CLS] token is then fed into an output layer for classification.

- **BART** [30] is the original transformer model that we build on by finetuning the pretrained model "bart-base-uncased". We concatenate the sentence pair separated by a  token, feeding it both into the encoder and the decoder. We use the  token at the end of the sentence pair for aggregate representation so that it can attend to decoder states from the complete input. The embedding for  is then fed into an output layer for classification.

- **RoBERTa** [34] shares the same transformer architecture with BERT, but uses a more robust pretraining strategy, and as a result performs better than BERT in many NLP tasks [34]. For our experiments, we employ RoBERTa in the same way as BERT.

- **IDP** (In-distribution Predictor) is our model for in-distribution prediction if we know a priori that the testing pairs come from the same distribution as the training pairs. It combines the positive and negative models as given in Eqn. 3.

- **OODP** (Out-of-distribution Predictor) is our model for out-of-distribution prediction if we know a priori that the testing and training pairs are not in the same distribution. It only makes use of the positive model. We expect our OODP to have better generalizability than our IDP, because of its reduced reliance on negative examples.

- **GAP** (Generalized Autoregressive Paraphrase-Identification) is our method that utilizes the perplexity based out-of-distribution detector to automatically control the reliance on the negative model, using Eqn. 4. This setting is different from IDP and OODP in that we do not have a priori knowledge of the test distribution.

- **GAPX** (Generalized Autoregressive Paraphrase-Identification X) ensembles GAP with RoBERTa described above, because we found RoBERTa to be the strongest baseline for in-distribution prediction. The intention is to capture via RoBERTa the interplay between positive and negative pairs. As depicted in Eqn. 8, when we do not have significant evidence that the given pair is out of distribution (p-test > 10%), we trust the prediction given by the discriminative model (RoBERTa). Otherwise, we trust GAP.

Following the previous literature in sentence matching [52, 16], we mainly use macro F1 score (F1) and accuracy score (ACC) to evaluate the models. Results based on Area-under-curve of the Receiver Operating Characteristic Curve (AUROC), a common metrics used to evaluate out-of-distribution metrics, are also provided in Appendix.A.

## 4.3 Measuring Distribution Shift

To situate our experiments properly, we note that different datasets does not equate to different distributions. It is thus important for us to be able to measure the distribution gap between datasets, and shed light on the models that perform the best when transferring between datasets with high distribution gap. Metrics that is per sample based such as our perplexity measure are not suitable for measuring at dataset level. To this end, we take a look at the Reverse Classification Accuracy* (RCA*) metric that has been proposed to predict the drop of model performance [18] and model selection [19, 64]. Here, given $D^s$ and $D^t$, we first train a model $M1$ from the training set of $D^s$. We then take a certain amount of samples from the target distribution $D^t$ (1000 in our experiments), and use $M1$ to relabel them. The relabeled pairs are then utilized to train a new model $M2$. We measure

the performance (in terms of AUC or ACC) of $M2$ on a held-out test set from $D^s$. As a control, we train another model $M3$ following the same procedure except the relabeled data comes from $D^s$. We denote the performance drop from $M3$ to $M2$ as the RCA* score indicative of the distribution gap.

To calibrate RCA*, for each dataset, we randomly selected 1000 held-out pairs and measure its shift from the dataset itself (which we expect to characterize an in-distribution RCA*). After 100 repetitions of measurements, we get a probability distribution of RCA* scores for each distribution in itself. Calibration results are reported in Appendix. 9. All the distributions share a mean of around 0 and a standard deviation around 2. By a p-test of 10%, it is unlikely that datasets with a RCA* greater than 4.0 would belong to the same distribution. Based on this observation, most of the transfers between different datasets are likely out-of-distribution, except for transfers from QQP to PIT and vice versa, which have a RCA* score of 2.8 and 3.4 respectively. This is probably because QQP and PIT are curated in a similar fashion, where they extract sentences with similar topics from social platforms, while all other datasets adopt different strategies to collect their data.

Finally, it is important to note that although RCA* provides a good estimate of the distribution shift at the dataset level, it's utility does not easily extend to Eqn. 4 and 8 as opposed to per sample out-of-distribution metrics. RCA* assumes availability of the entire test set, while in real world, we are much more likely to get a single or small batches of test pairs at a time, all of which could even be from different distributions.

## 4.4 Implementation details

In practice, for testing, it helps to average the cross entropy by the length of the sentence, and average the cross entropy of generating $s_2$ from $s_1$ and generating $s_1$ from $s_2$. To optimize conditional sentence generators, we use Adam optimizer with learning rate 2e-5. We adopt cross entropy loss for each word logit. All experiments are run on Nvidia 2080 Ti with 11 GB memory.

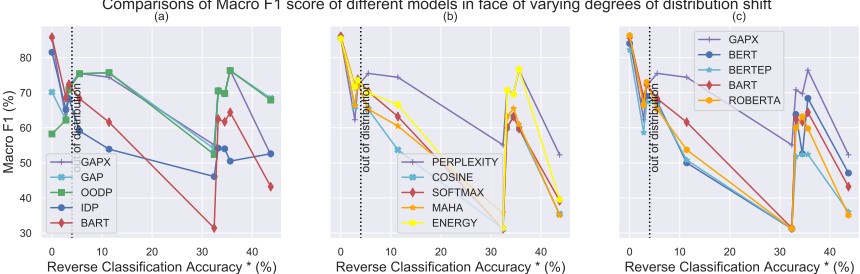

Figure 3: Comparing Macro F1 scores of different models at varying degrees of distribution shift.

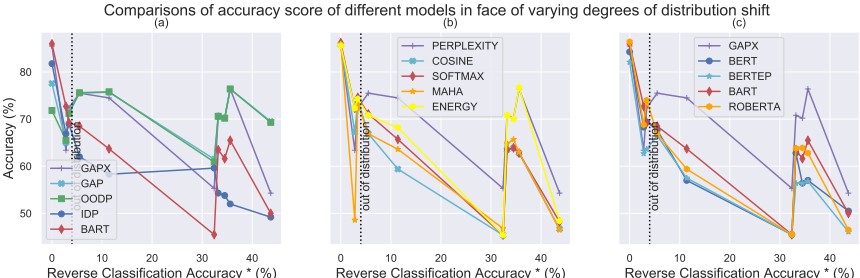

Figure 4: Comparing ACC score of different models at varying degrees of distribution shift.

## 4.5 Main Results

**Bias in Negative Pairs**    To understand whether negative pairs are major sources of bias, we plot the Macro F1 and ACC score against RCA* in Fig. 3(a) and Fig. 4(a), comparing the performances of OODP, IDP, and BART, all finetuned from the same pretrained checkpoint. The x-axis is plotted in ascending order of RCA* between pairs of datasets given in Table 1 and 2. There are three pairs

| Model | QQP ->WMT (5.5) | PIT ->WMT (11.4) | PIT ->PAWS (32.4) | PAWS -> QQP (33.2) | PAWS -> PIT (34.5) | PAWS -> WMT (35.6) | QQP -> PAWS (43.7) | average |
|---|---|---|---|---|---|---|---|---|
| BOW | 34.6/51.5 | 33.3/51.4 | 35.3/54.7 | 33.3/50.0 | 34.3/50.2 | 34.8/51.7 | 35.3/54.7 | 34.4/52.0 |
| BiLSTM | 34.4/51.5 | 50.7/51.1 | 48.6/48.7 | 36.8/50.4 | 43.3/50.6 | 34.9/51.2 | 37.1/54.7 | 40.8/51.2 |
| BERT | 67.4/67.7 | 50.0/57.7 | 31.2/45.5 | 63.8 /62.8 | 52.6/56.4 | 68.4/57.0 | 47.1/50.5 | 54.4/56.8 |
| BERT+EP | 66.5/66.5 | 50.8/57.5 | 31.2/45.5 | 51.8/56.4 | 52.4/56.4 | 52.4/56.8 | 36.0/46.1 | 48.7/55.0 |
| BART | 68.3/68.5 | 61.6/63.7 | 31.4/45.5 | 62.5/63.5 | 61.7/61.6 | 64.4/65.5 | 43.2/50.0 | 56.2/59.8 |
| RoBERTa | 65.3/66.9 | 53.7/59.4 | 31.2/45.5 | 60.0/63.6 | 63.2/63.9 | 59.8/62.8 | 35.1/46.5 | 52.6/58.4 |
| IDP | 59.1/62.0 | 53.9/58.3 | 46.1/59.6 | 54.2 /54.3 | 54.0/53.8 | 50.5/52.0 | 52.6/49.2 | 52.9/55.6 |
| OODP | 75.4/**75.6** | **75.7/75.8** | 52.4/60.9 | 70.5/70.6 | **69.8/70.2** | 76.3/**76.4** | 67.9/69.3 | 69.7/**71.3** |
| GAP | 75.4/**75.6** | **75.7**/75.7 | 54.0/**61.5** | 70.5/70.6 | 69.7/**70.2** | 76.3/**76.4** | **68.4/69.5** | 70.0/71.3 |
| GAPX | **75.5**/75.5 | 74.4/74.5 | **55.1**/55.5 | **70.8/70.8** | 69.7/**70.2** | **76.4/76.4** | 52.3/54.3 | 67.7/68.2 |

Table 1: Model performance on different out-of-distribution combinations of QQP, PAWS and PIT, in terms of macro F1/accuracy (ACC). Parenthesized is the RCA* score for each combination of datasets.

| Model | QQP ->QQP (0) | PAWS ->PAWS (0) | PIT ->PIT (0) | QQP ->PIT (2.8) | PIT ->QQP (3.4) | average |
|---|---|---|---|---|---|---|
| BOW | 51.3/57.8 | 48.8/56.8 | 33.3/50.0 | 41.7/49.7 | 33.3/50.0 | 40.7/52.9 |
| BiLSTM | 61.6/63.6 | 43.6/53.0 | 50.6/51.1 | 51.7/52.9 | 41.1/49.0 | 49.7/53.9 |
| BERT | 82.5/82.6 | 92.7/93.2 | 76.9/77.0 | 68.0/68.3 | 69.0/69.4 | 77.8/78.1 |
| BERT+EP | 81.6/81.7 | 89.7/89.7 | 75.3/74.9 | 58.6/62.7 | 67.4/63.6 | 74.5/74.5 |
| BART | 82.6/82.8 | 94.1/94.1 | 80.9/81.0 | 68.6/72.7 | 72.4/69.2 | 79.7/80.0 |
| RoBERTa | 84.4/84.5 | 93.5/93.6 | 81.0/81.1 | 66.5/68.8 | 73.0/74.9 | 79.7/80.6 |
| OODP | 65.3/73.2 | 67.9/77.1 | 41.5/65.2 | 62.2/65.5 | 65.2/71.2 | 60.4/70.4 |
| IDP | 79.0/79.1 | 88.2/88.5 | 77.4/77.7 | 65.2/66.9 | 68.3/69.0 | 75.6/76.2 |
| GAP | 68.8/71.0 | 85.1/85.2 | 56.6/76.5 | 62.2/65.0 | 71.3/71.7 | 68.8/73.9 |
| GAPX | 84.4/84.5 | 92.7/92.7 | 79.3/79.3 | 62.3/63.4 | 72.0/72.4 | 78.1/78.5 |

Table 2: Model performance for in-distribution performances on QQP, PAWS, and PIT, in terms of macro F1/accuracy (ACC). Parenthesized is the RCA* score for each combination of datasets.

that have RCA* score of 0 (Table 2), for which we average the performance in the plots. Both the F1 and ACC plots share a similar pattern. In the in-distribution region, where the RCA* score is less than 4% (Fig. 9), BART and IDP achieves similarly high performances of 79.7% and 75.6% F1 on average respectively. The 4.1% gap in F1 is potentially due to the fact that IDP trains a negative model and positive model separately, neglecting the interaction between positive and negative pairs. Comparatively, the performance of OODP is significantly inferior to the other two, with only 60.4% average F1. This changes in the out-of-distribution region, where the RCA* score is now greater than 4%. OODP turns out to be the leading model over BART and IDP. With increasing degree of distribution shift, we observe that both BART and IDP are especially fragile and their performance drop significantly. When the RCA* is greater than 20, both models' F1 drop to as low as around 60%, which can be hardly useful in practice. In contrast, OODP maintains an advantage in F1 of as much as 10-20% throughout the out-of-distribution region. Since the only difference between OODP and IDP is that OODP transfers only the positive model, it confirms our hypothesis that the negative model does not generalize as well as the positive model.

**Importance of the Interplay between Positive and Negative Pairs**    To understand the necessity of capturing the interplay for test pairs that are in distribution, we compare the performances of GAP, GAPX, and BART in Fig. 3(a) and Fig. 4(a). GAP only ensembles the positive and negative model trained separately, so does not contain any interplay information. On the other hand, both BART, trained with positive and negative pairs together, and GAPX (Sec. 3.2.2) capture interplay information. As shown in the plot, the performance of GAP in the in-distribution region is not directly comparable to BART with a gap of 10.9% in macro F1. In contrast, GAPX's in-distribution result has a much smaller margin of 1.6% macro F1 compared to BART, yet, by automatically weighing the contribution of the discriminative model, GAPX also closely matches the performance of GAP in the out-of-distribution region with only a 2.3% loss in macro F1.

**Effectiveness of Perplexity-based Ensembling**   We also substitute $PP(s_1, s_2)$ with other state-of-the-art out-of-distribution metrics used for estimating the probability in Eqn. 6 for GAPX. Specifically, we compare our perplexity metric with Maximum Softmax Probability (SOFTMAX) [24, 26, 6], Energy Score (ENERGY) [33], Mahalanobis Distance (MAHA) [29], and COSINE [65].  See Appendix for more details. The results are given in Fig. 4(b) and Fig. 3(b). F1 and ACC are all similar in the in-distribution region. However, in the out-of-distribution region, SOFTMAX, MAHA, and COSINE start to perform poorly. ENERGY turned out to be the most robust but is still obviously not matching our perplexity metric, often by a large margin.

**Generalization**   We implemented one of the most popular methods for domain generalization in paraphrase identification, Expert Product [15], as a potential strong baseline for handling dataset bias.  As described in Sec. 4.2, we train a BERT classifier with only the first sentence $s_1$ as the biased model for BERT+EP. We report the results in Fig. 4(c) and Fig. 5(c), together with BERT, BART and RoBERTa. In addition, we also provide performance of traditional methods like BoW [62] and ESIM [12] in Table 1 and Table 2.  BERT, BART, and RoBERTa all produce similar results on all combinations of datasets. We observe that their performances are consistently better than traditional methods like BoW and ESIM, showing that pretraining and finetuning can indeed improve the generalizability of classifiers for paraphrase identification. However, their performances in out-of-distribution setting are still far from their in-distribution performances, with accuracy below 65% in most of the cases (around 20% drop). GAPX maintains an absolute margin of around 10% in terms of ACC and an absolute margin of 7-20% in terms of F1 in the out-of-distribution region. The best transformer-based models in the out-of-distribution region is BART, with an average of 56.2% in F1 and 59.8% in ACC, while GAPX maintains an average of 67.7% in F1 and 68.2% in ACC, with an absolute gain of 11.5% in F1 and 8.4% in ACC. What is also encouraging is that GAPX's performance is close to that of OODP, which is promising that we do not need *a priori* information on the domain gap between the source and target. Lastly, BERT+EP fails to provide much gain, which we conjecture is due to the difficulty of "finding" the right bias model or features.

## 5   Conclusion

We have shown that negative samples introduce bias that prevent the generalization of paraphrase identification models. To overcome, we present a novel paradigm to train separate models for the distribution of positive and negative samples independently, and utilize a perplexity based out-of-distribution detection to ensemble them automatically. Experiments show that our method achieves an average of 11.5% gain of F1 and 8.4% gain of ACC in various different out-of-distribution scenarios over other state-of-the-art methods.

### 5.1   Limitations

Our methodology is specifically designed for only "verification" problems, where the samples come in pairs. Scenarios that involve other types of bias will require non-trivial turn-key formulations to explicitly model the source of bias (much like the negative model).

## 6   Acknowledgements

This work is sponsored by Meta AI. We would also like to thank Prof. Yoav Artzi for a helpful discussion in the early stage of this project.

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
