# A AUROC of Main Results

We include full results of AUROC scores here.

| Model | QQP -> WMT (5.5) | PIT -> WMT (11.4) | PIT -> PAWS (32.4) | PAWS -> QQP (33.2) | PAWS -> PIT (34.5) | PAWS -> WMT (35.6) | QQP -> PAWS (43.7) | average |
|---|---|---|---|---|---|---|---|---|
| BOW | 51.9 | 52.5 | 49.0 | 56.1 | 44.0 | 53.2 | 49.1 | 50.8 |
| BiLSTM | 50.1 | 51.5 | 49.3 | 50.4 | 50.6 | 49.8 | 50.1 | 50.3 |
| BERT | 75.0 | 70.7 | 51.2 | 69.2 | 58.2 | 70.7 | 54.7 | 64.2 |
| BERT+EP | 73.5 | 74.6 | 52.2 | 66.9 | 62.3 | 69.4 | 53.4 | 64.6 |
| BART | 75.7 | 76.4 | 53.3 | 71.7 | 65.0 | 77.6 | 56.6 | 68.0 |
| RoBERTa | 77.9 | 76.9 | 52.4 | 77.4 | 71.2 | 80.6 | 54.3 | 70.1 |
| IDP | 71.0 | 65.7 | 63.9 | 56.0 | 55.0 | 50.3 | 67.0 | 61.3 |
| OODP | 85.0 | **85.0** | **67.2** | 76.7 | **77.5** | **84.9** | 74.4 | **78.7** |
| GAP | **85.1** | 84.7 | 66.4 | 76.2 | **77.5** | **84.9** | **74.7** | 78.5 |
| GAPX | 83.8 | 81.1 | 58.3 | **77.7** | 77.5 | 84.6 | 59.5 | 74.6 |

Table 3: Model performance on different out-of-distribution combinations of QQP, PAWS and PIT, in terms of area under curve (AUROC).

| Model | QQP ->QQP (0) | PAWS ->PAWS (0) | PIT ->PIT (0) | QQP ->PIT (2.8) | PIT ->QQP (3.4) | average |
|---|---|---|---|---|---|---|
| BOW | 60.9 | 57.7 | 47.7 | 47.4 | 61.3 | 55.0 |
| BiLSTM | 64.6 | 49.6 | 50.9 | 52.8 | 49.0 | 53.4 |
| BERT | 90.0 | 97.6 | 85.1 | 77.3 | 73.4 | 84.7 |
| BERT+EP | 89.8 | 95.8 | 82.7 | 76.1 | 73.7 | 83.6 |
| BART | 90.6 | 98.2 | 89.1 | 78.2 | 77.9 | 86.8 |
| RoBERTa | 92.0 | 98.7 | 90.2 | 82.0 | 80.0 | 88.6 |
| OODP | 79.6 | 84.6 | 73.6 | 72.8 | 76.4 | 77.4 |
| IDP | 88.1 | 95.4 | 86.8 | 77.9 | 74.6 | 84.6 |
| GAP | 78.9 | 92.3 | 84.5 | 72.3 | 76.4 | 80.9 |
| GAPX | 90.7 | 98.1 | 87.4 | 72.5 | 79.2 | 85.6 |

Table 4: Model performance for in-distribution performances on QQP, PAWS, and PIT, in terms of area under curve (AUROC).

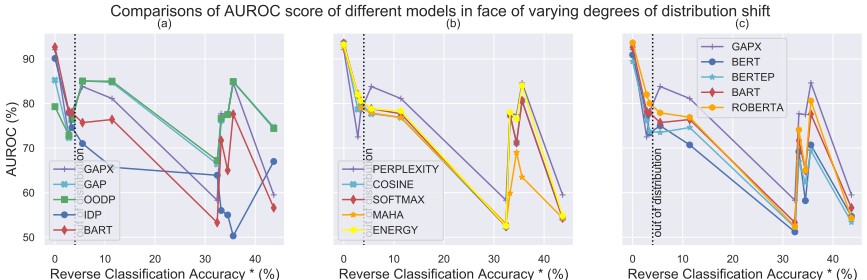

Figure 5: Comparing AUROC scores of different models at varying degrees of distribution shift.

# B Additional Ablations

## B.1 Additional Variants

We also conducted ablations on several variants of GAPX.

- **GAPX(neg-log)** replaces perplexity with the neg-log likelihood of the concatenated $(s_1, s_2)$ as the measure. Specifically, GAPX(neg-log) modifies the Eqn. 5 and Eqn. 6 such that:

$$NLL(s_1, s_2) = \log((\prod_{i=1}^{n} \frac{1}{P(w^i|w^{1:i-1})}).$$

$$\lambda(s_1, s_2) = cdf(NLL(s_1, s_2), Weibull(a, c, loc, scale)).$$

- **GAPX(w/ IDP)** ensembles GAP with IDP.

- **GAPX(w/ BART)** explores the option of ensembling GAP with BART as described in Appendix.4.2.

These variants of GAPX mostly perform similarly to the GAPX model proposed in the main paper, which ensembles GAP with RoBERTa.

| Model | QQP -> WMT (5.5) | PIT -> WMT (11.4) | PIT -> PAWS (32.4) | PAWS -> QQP (33.2) | PAWS -> PIT (34.5) | PAWS -> WMT (35.6) | QQP -> PAWS (43.7) | average |
|---|---|---|---|---|---|---|---|---|
| GAPX | 75.5/75.5 | 74.4/74.5 | 55.1/55.5 | 70.8/70.8 | 69.7/70.2 | 76.4/76.4 | 52.3/54.3 | 67.7/68.2 |
| GAPX(neg-log) | 75.9/75.9 | 74.8/74.8 | 56.0/61.7 | 62.6/65.9 | 62.9/63.4 | 72.3/72.4 | 62.9/63.0 | 66.8/68.2 |
| GAP(w/ IDP) | 73.2/73.4 | 74.0/74.0 | 55.4/61.6 | 57.8/58.3 | 54.7/54.7 | 67.7/67.9 | 66.3/66.5 | 64.2/65.2 |
| GAPX(w/ BART) | 74.6/74.6 | 74.6/74.6 | 55.2/55.4 | 70.7/70.8 | 69.7/70.2 | 75.9/76.0 | 55.1/56.0 | 68.0/68.2 |
| GAPX(w/ BERT) | 72.6/72.6 | 74.7/74.7 | 55.1/55.2 | 70.2/70.2 | 69.7/70.2 | 75.7/75.9 | 56.1/56.3 | 67.7/67.9 |

Table 5: Performance on different out-of-distribution combinations of QQP, PAWS and PIT, in terms of macro F1/accuracy (ACC).

| Model | QQP ->QQP (0) | PAWS ->PAWS (0) | PIT ->PIT (0) | QQP ->PIT (2.8) | PIT ->QQP (3.4) | average |
|---|---|---|---|---|---|---|
| GAPX | 84.4/84.5 | 92.7/92.7 | 79.3/79.3 | 62.3/63.4 | 72.0/72.4 | 78.1/78.5 |
| GAPX(neg-log) | 83.3/83.4 | 91.2/91.5 | 80.9/80.9 | 68.9/70.1 | 72.2/72.7 | 79.3/78.7 |
| GAP(threshed) | 78.2/78.3 | 86.5/86.8 | 77.4/77.7 | 67.4/68.5 | 69.4/69.8 | 75.8/76.2 |
| GAPX(w/ BART) | 82.6/82.8 | 93.0/93.1 | 79.4/79.4 | 62.3/63.4 | 71.4/71.5 | 77.7/78.0 |
| GAPX(w/ BERT) | 82.6/82.7 | 91.9/92.0 | 76.8/76.8 | 62.3/63.4 | 69.9/69.9 | 76.7/77.0 |

Table 6: In-distribution performances on QQP, PAWS, and PIT, in terms of macro F1/accuracy (ACC).

| Model | QQP -> WMT (5.5) | PIT -> WMT (11.4) | PIT -> PAWS (32.4) | PAWS -> QQP (33.2) | PAWS -> PIT (34.5) | PAWS -> WMT (35.6) | QQP -> PAWS (43.7) | average |
|---|---|---|---|---|---|---|---|---|
| GAPX(perplexity) | 75.5/75.5 | 74.4/74.5 | 55.1/55.5 | 70.8/70.8 | 69.7/70.2 | 76.4/76.4 | 52.3/54.3 | 67.7/68.2 |
| GAPX(cosine) | 65.2/66.9 | 53.7/59.4 | 31.2/45.4 | 60.0/63.5 | 63.2/63.9 | 59.8/62.8 | 35.4/46.7 | 52.7/58.4 |
| GAPX(softmax) | 70.3/71.1 | 63.2/65.7 | 31.2/45.4 | 60.0/63.5 | 63.2/63.9 | 59.8/62.8 | 39.2/48.2 | 55.3/60.1 |
| GAPX(maha) | 65.3/66.9 | 60.5/63.6 | 35.7/46.8 | 63.2/64.8 | 65.5/65.7 | 61.0/63.0 | 35.1/46.5 | 55.2/59.6 |
| GAPX(energy) | 69.9/70.8 | 66.6/68.2 | 31.4/45.4 | 70.7/70.8 | 69.5/70.0 | 76.7/76.7 | 39.5/48.4 | 60.6/64.3 |

Table 7: Out-of-distribution performance when using different out-of-distribution metrics.

| Model | QQP ->QQP (0) | PAWS ->PAWS (0) | PIT ->PIT (0) | QQP ->PIT (2.8) | PIT ->QQP (3.4) | average |
|---|---|---|---|---|---|---|
| GAPX(perplexity) | 84.4/84.5 | 92.7/92.7 | 79.3/79.3 | 62.3/63.4 | 72.0/72.4 | 78.1/78.5 |
| GAPX(cosine) | 82.6/82.8 | 93.5/93.5 | 81.0/81.1 | 66.0/67.2 | 73.0/74.0 | 79.2/79.7 |
| GAPX(softmax) | 82.2/82.5 | 93.5/93.5 | 82.6/82.6 | 71.9/72.4 | 73.8/74.6 | 80.8/81.1 |
| GAPX(maha) | 84.4/84.5 | 92.2/92.2 | 81.3/81.3 | 66.4/68.6 | 72.5/73.2 | 79.4/80.0 |
| GAPX(energy) | 82.8/83.0 | 92.2/92.2 | 81.4/81.5 | 71.7/72.3 | 73.6/74.3 | 80.3/80.7 |

Table 8: In-distribution performance of different out-of-distribution metrics.

# C Additional Discussions

## C.1 Ablations on $C$, Eqn. 4

| Model | QQP -> PIT (2.8) | PIT -> QQP (3.4) | PIT -> PAWS (32.4) | PAWS -> QQP (33.2) | PAWS -> PIT (34.5) | QQP -> PAWS (43.7) | average |
|---|---|---|---|---|---|---|---|
| GAPX(0) | 62.3/63.4 | 72.0/72.4 | 55.1/55.5 | 70.8/70.8 | 69.7/70.2 | 52.3/54.3 | 63.7/64.4 |
| GAPX(10) | 63.3/64.0 | 72.1/72.7 | 48.3/51.5 | 69.5/69.5 | 69.7/70.2 | 47.3/52.4 | 61.7/63.4 |
| GAPX(100) | 66.9/67.1 | 72.1/72.6 | 48.4/50.7 | 71.3/71.5 | 71.2/71.3 | 52.3/54.4 | 63.7/64.6 |

Table 9: Comparing using different amount of data to set $C$. Macro F1/ACC are reported.

To set $C$ in Equation. 4, in the case of 0 samples, we roughly estimate an integer from 1-5 for the interpolation constant $C$. For adversarial distributions like PAWS, we expect a lower perplexity because both sentences share the same bag-of-word, so we set $C = 1$. For standard distributions like QQP and WMT, we expect a modest perplexity, so we set $C = 3$. For informal distributions like PIT, where sentences do not strictly follow syntax and grammar, we expect a higher perplexity, so we set $C = 5$. In other cases, with validation data, we determine the best constant $C$ based on the validation data (if there are multiple constants with the same results on the validation data, we take the smallest one). As shown in Table 3 (WMT results are not included here due to the lack of validation data), although using 0 or 10 samples achieve worse performances than using 100 samples, meaning the best constant threshold is not found, the results appear to be stable overall (on average 2% fluctuations in macro F1 and 1% fluctuations in accuracy). If a small amount of validation data is accessible, the performance of GAPX can be further improved.

## C.2 Interpreting the Results

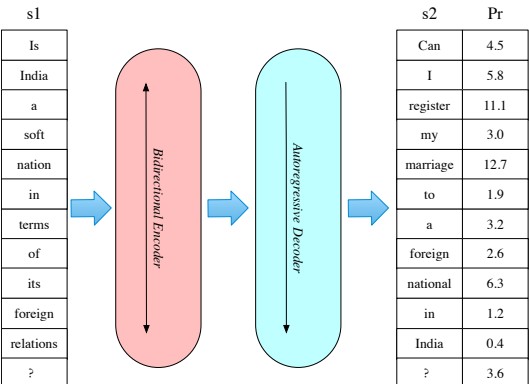

Figure 6: An example from QQP illustrating how to interpret the result of our method, by OODP.

Figure 6, Figure 7, and Figure 8 shows examples of how our autoregressive paraphrase identification models work. For OODP, our model will output a log of conditional probability for each word in $s_2$ given $s_1$ and all the previous words in $s_2$, namely:

$$\log P(w_2^{(i)}|s_1, Y = 1, w_2^{(1:i-1)}).$$

For IDP, we can use the log of the quotient of the conditional probability for each word given by the positive model and the negative model as an indicator which words contribute the most to the prediction result:

$$\log P(w_2^{(i)}|s_1, Y = 1, w_2^{(1:i-1)}) - \log P(w_2^{(i)}|s_1, Y = 0, w_2^{(1:i-1)}).$$

For GAP, we can use the the score defined in Eqn.. 4 split on each word, namely:

$$\log P(w_2^{(i)}|s_1, Y = 1, w_2^{(1:i-1)}) - (1 - \lambda(s_1, s_2)) \log P(w_2^{(i)}|s_1, Y = 0, w_2^{(1:i-1)}) - \lambda(s_1, s_2)C.$$

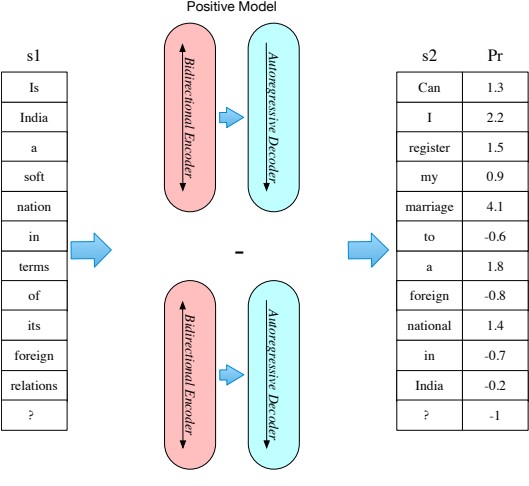

Figure 7: An example from QQP illustrating how to interpret the result of our method, by IDP.

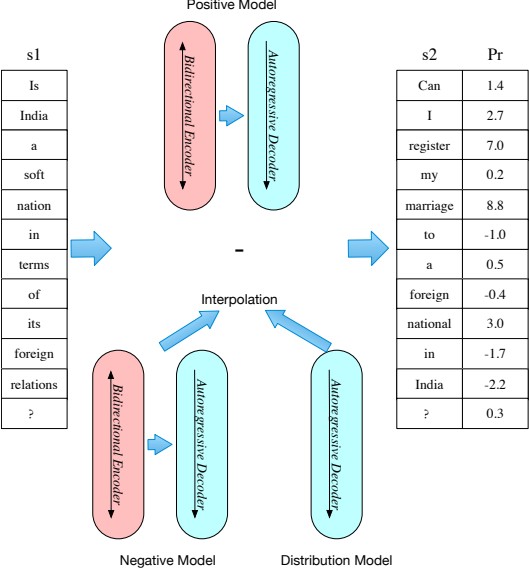

Figure 8: An example from QQP illustrating how to interpret the result of our method, by GAP.

In all three models, higher scores represent a higher chance of being non-paraphrases. For IDP and GAP, the threshold is 0 while for OODP the threshold is 3. All three models predict this sentence pair to be non-paraphrases, attending to slightly different key words. The top 3 words with the highest scores in OODP and GAP are 'register', 'marriage', and 'national'. All of them represent the words that are unlikely to occur in a paraphrase of the original sentence. The top 3 words with the highest scores in IDP are 'I', 'marriage', and 'a'. Its reliance on the word 'a' might be due to the error propagation of the autoregressive decoding.

## C.3 Implementation of out-of-distribution metrics

To compare our perplexity metric with different off-the-shelf out-of-distribution metrics (Fig. 4(b) and 5(b)), we first train a RoBERTa model as described in Sec. 4.2. Both MAHA and COSINE need in-distribution validation data, for which we use half of the development data provided in each dataset. We use the other half of the development data to estimate the Weibull distribution of the

metrics. We rely on the implementation of SOFTMAX, ENERGY, MAHA, and COSINE from Zhou and Chen [65]. The metrics are calculated as follow:

1. **SOFTMAX**. We use the maximum class probability $1 - max_{j=0,1}p_j$ among 2 classes (paraphrases and non-paraphrases) in the final softmax layer.

2. **ENERGY**. We use the following formula to calculate energy score:
$$g = -\log \sum_{j=0}^{1} \exp\left(w_j^T h\right),$$
where $w_j$ is the weight of the $j$th class in the softmax layer, and $h$ is the input to the softmax layer (of the concatenated $(s_1, s_2)$ input).

3. **MAHA**. We use the input representation $h$ of the penultimate layer of the model, and fit a Gaussian distribution to each class in the in-distribution development data $\mathcal{D}_{val} = \{(x_i, y_i)\}_{i=1}^{M}$:
$$\mu_j = \mathbf{E}_{y_i=j}[h_j]$$
$$\Sigma = \mathbf{E}[(h_i - \mu_{y_i})(h_i - \mu_{y_i})^T].$$

Then, the MAHA distance is calculated as:
$$g = -\min_{j=0,1}(h - \mu_j)\Sigma^+(h - \mu_j),$$
where $\Sigma^+$ is the pseudo-inverse of $\Sigma$.

4. **COSINE**. We use the maximum cosine similarity of $h$ (of the concatenated $(s_1, s_2)$ input) to samples in the validation dataset:
$$g = -\max_{i=1}^{M} \cos\left(h, h_i^{(val)}\right).$$

### C.4 RCA Implementation Details

To instantiate a measurement of RCA* scores. We use the RoBERTa model described in Section 4.2 as our classifier. To measure the distribution shift from a source distribution $\mathcal{D}^s$ to a target distribution $\mathcal{D}^t$. We use a training set of $\mathcal{D}^s$, a test set of $\mathcal{D}^s$, and a development set of $\mathcal{D}^t$. All the measurements of RCA* scores in this paper fix the size of the development to be 1000, and use the test set of $\mathcal{D}^s$ to be the entire test set of the original dataset.

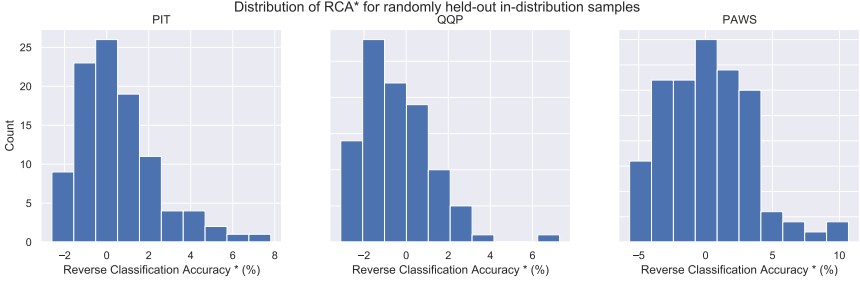

Figure 9: Distribution of RCA* for randomly held-out in-distribution samples

To calibrate the RCA* scores, we measure the distribution shift of one dataset to itself. We first train a binary classifier RoBERTa $M1$ (of paraphrases and non-paraphrase) with the training set of $\mathcal{D}^s$ for 3 epochs with an Adam Optimizer of learning rate 2e-5 (training with multiple runs with different random seeds to select the best model on a development set of $\mathcal{D}^s$, different from the development set of $\mathcal{D}^t$). Then we apply $M1$ to the development set of $\mathcal{D}^t$ to relabel those data. We take the relabeled data to retrain a classifier $M2$ and apply $M2$ to the test set of $\mathcal{D}^s$. We measure the performance drop from $M1$ to $M2$ on the test set of $\mathcal{D}^s$ as the RCA score. Note here that we make small changes to the originally proposed RCA score where the performances are measured in terms of ACC scores. We found that AUROC scores are in practice more stable for measuring RCA scores, so we use AUROC scores instead. To get the final RCA* score, we use the equation:
$$RCA * (\mathcal{D}^s, \mathcal{D}^t) = RCA(\mathcal{D}^s, \mathcal{D}^t) - RCA(\mathcal{D}^s, \mathcal{D}^s),$$

where $RCA*(\mathcal{D}^s, \mathcal{D}^t)$ is the RCA* score from $\mathcal{D}^s$ to $\mathcal{D}^t$, while $RCA(\mathcal{D}^s, \mathcal{D}^t)$ is the RCA score from $\mathcal{D}^s$ to $\mathcal{D}^t$. Under this definition, the RCA* score of one dataset to itself is defined to be 0. To repeat the measurements of distribution shift from $\mathcal{D}^s$ to itself, at each time, we hold-back 1000 random pairs from $\mathcal{D}^s$, and measure the RCA. We include plots of the distribution of the RCA* in Fig. 9 as well as the corresponding raw RCA* data below:

PIT: 4.4, 0.65, 1.9, 0.91, 5.1, 4.0, 1.1, 0.63, -0.1, -0.43, 2.9, 0.6, -1.7, 0.83, -0.27, 7.8, 1.7, 3.8, 0.78, -1.2, 2.1, 1.1, -1.3, -1.2, 1.3, -1.2, 0.032, 3.0, -2.0, -1.8, -1.2, 1.8, -2.4, 6.2, -0.47, 0.26, -0.88, -1.1, 4.2, 3.4, 3.3, 0.22, -0.25, 0.065, 2.6, 0.15, -0.93, 0.27, -0.49, -1.5, -0.79, 0.38, 2.3, 0.83, -1.4, 0.81, -0.63, -1.6, 0.97, -1.6, 0.82, 1.1, 1.9, 2.3, 0.23, -1.1, 0.72, -0.87, 1.7, 0.04, -0.38, 0.23, 1.7, -1.2, 0.1, 1.4, -0.15, 1.2, -2.6, -1.4, 0.75, 2.3, -1.2, -0.68, 5.6, -0.13, -1.1, 0.16, -0.39, 0.097, -1.9, -1.1, -0.26, -1.3, -1.3, -0.8, 0.3, 0.73, -1.9, -0.51

QQP: -0.27, 0.72, -2.3, 1.3, -2.5, 1.1, -1.1, -1.4, 2.0, -2.1, -0.77, -0.63, 0.045, -0.97, -2.1, 0.11, 1.3, -0.25, -1.1, 0.39, 0.65, 1.2, 0.87, 1.5, 0.058, 3.5, -0.039, 1.6, -1.2, 2.5, 0.64, -0.48, 2.5, -1.8, -0.52, -2.1, -0.22, -1.4, 0.97, 3.0, -2.9, -0.92, -0.42, 0.72, 1.8, -1.3, 0.63, -2.0, 0.4, -1.2, -0.65, -2.0, 0.7, 0.95, -1.3, -2.1, 1.2, -1.0, 2.5, -0.32, -1.8, -0.59, -0.016, -1.4, -1.3, 2.5, 1.4, -2.2, -1.2, 0.98, 0.93, 0.98, -2.1, -1.1, -1.8, -0.43, -0.42, 7.3, -0.75, -1.5, -0.87, -2.4, -0.61, 0.084, -1.7, -0.16, -2.4, -2.0, -1.6, -2.9, -1.7, -2.3, -1.9, -2.3, -1.7, -3.1, -1.6, 0.2, -1.7, -2.9

PAWS: -3.0, -0.0092, -0.46, -5.6, -1.7, 2.7, 0.5, 2.4, 3.6, 0.034, -4.1, -3.2, 3.6, 8.2, 0.71, -0.079, 2.9, 5.4, 1.7, -5.2, 3.2, 2.8, -3.0, 2.0, -2.2, -1.7, -0.2, 4.1, -0.22, 3.9, 1.1e+01, 1e+01, 1.2, 0.0074, -0.91, -2.2, 3.9, -2.9, -2.7, -2.8, -4.2, -4.1, -1.0, -3.1, 2.6, -0.83, 0.74, 1.7, 0.56, 2.2, 0.87, -0.5, 0.83, -5.5, 0.4, -3.0, 1.1, -3.1, 0.92, -1.6, 4.1, -3.1, -0.99, 0.79, -0.15, -1.3, 5.2, 0.18, -5.7, 3.4, 2.9, 3.2, -4.0, -2.4, -4.9, -0.51, -1.8, -2.7, -3.7, 6.6, 0.65, 2.5, 1.5, -2.3, 3.1, 1.9, 1.2, -2.1, 1.7, 0.95, 2.1, -2.7, 5.9, -4.0, 1.8, -1.8, -0.2, 4.6, -1.3, -1.0