# OpenReview forum: "GAPX: Generalized Autoregressive Paraphrase-Identification X"
_NeurIPS.cc/2022/Conference — NeurIPS 2022 Accept_

### Official Review · Reviewer_KXFG · 2022-07-09

**Rating:** 8
**Confidence:** 4
**Soundness:** 4 excellent
**Presentation:** 4 excellent
**Contribution:** 3 good

**Summary:**

**What is the task?**
Paraphrase identification

**What has been done before?**
* Many state-of- the-art models often suffer from distribution shift during inference time. Author(s) show a major source of this performance drop comes from biases introduced by negative examples.To overcome these biases, they propose in this paper to train two separate models, one that only utilizes the positive pairs and the other the negative pairs.

* Authors have compared their work with different lines of work like distribution shift and debiasing models in NLP, out of distribution detection, text generation metrics etc. and are able to show the novelty of their work.


**What are the main contributions of the paper?**
* Reported a new research insight, supported by empirical results, that the negative pairs of a dataset could potentially introduce biases that will prevent a paraphrase identification model from generalizing to out-of-distribution pairs.

* Proposed a novel autoregressive modeling approach to train both a positive and a negative model, and ensemble them automatically during inference.

* Introduced a new perplexity based approach to determine whether a given pair is out-of-distribution to achieve auto ensembling.

* SOTA results in out-of-distribution performance while keeping comparable performance for in-distribution prediction.



**What are the main results? Are they significant?**
Author(s) support their findings with strong empirical results using following experiments

* (1) verify that the task of paraphrase identification suffers from biases in the datasets that is the main obstacle to generalization in this field of study,
* (2) test the accuracy of proposed perplexity based out-of-distribution detection method, and
* (3) test that balancing the utilization of the negative model can help outperform the state-of-the-art in the face of distribution shift.


**Questions:**

NA

**Limitations:**

Authors have adequately addressed the limitations and potential negative societal impact of their work.

**Strengths And Weaknesses:**


Strengths

* Reported new research insights well supported by empirical results for all the findings like "bias in negative pairs", "importance of the interplay between positive and negative pairs", "effectiveness of perplexity-based ensembling", "generalization" etc.
* Good presentation - Paper is easy to understand
* SOTA results in out-of-distribution performance while keeping comparable performance for in-distribution prediction.

---

> ### Author Response · Authors · 2022-07-28
> **Thank you for your strong support**
>
>
> We really appreciate this reviewer's thorough understanding of our paper and its research and scientific value.

---

### Official Review · Reviewer_emd5 · 2022-07-11

**Rating:** 6
**Confidence:** 4
**Soundness:** 2 fair
**Presentation:** 2 fair
**Contribution:** 3 good

**Summary:**

This paper focuses on the distribution shift problem in paraphrase
identification task, and proposes several methods to better deal with
the bias brought by negative examples, including training two
auto-regressive models exclusively and respectively on positive and
negative examples, combining them with an ordinary discriminative
model, and determining their weights automatically on the basis of
average token-level perplexity.  Through a cross-dataset experiment,
the authors confirm that the proposed method is less affected by the
distribution shift of negative examples, while achieving a competitive
classification accuracy in the in-domain settings.  One advantage of
the proposed method is that it does not require us to have a knowledge
of the degree of distribution gap between training and test data a
priori.

**Questions:**

I have several questions regarding the proposed method.

- Considering the diversity of negative examples, The distribution learned from a limited amount of negative examples would not be smooth enough.  Is there any justification to apply an auto-regressive model, a kind of generative model, rather than instance-based computation of the likelihood, such as nearest neighbors?

- Two $P(s_{2}|s_{1},Y)$ in GAP (Eq.(4)) are not defined.  Are they respectively equal to the $\sum\log P()$ in IDP (Eq.(3))?

- When computing the perplexity scores, did the authors concatenate given a pair of $s_{1}$ and $s_{2}$ without any special tokens or not?  If not, how can the model properly evaluate the adjacency of the last token of $s_{1}$ and the first token of $s_{2}$?  What is the advantage of concatenating it over separately scoring for $s_{1}$ and $s_{2}$?

- The reason to choose the Weibull distribution and its parameters are unclear.

- How large was $M$ in GAPX (Eq.(8))?  It is explained as "a sufficiently large constant" but this means that the second term (GAP) will be ignored.

- There is $(\frac{1}{2} - P())$ but isn't it $(P() - \frac{1}{2})$?

**Limitations:**

Yes.  The proposed method exploits neural generative models but their usage is limited to labeling given pairs of sentences.  The trained models will not introduce any societal influence nor misleading conclusions.


**Strengths And Weaknesses:**

There are two noteworthy strengths.

- It empirically demonstrates that the performance of existing methods
can dramatically worsen when the model is applied to the dataset that
exhibits different distribution from the training data.

- The proposed method automatically determines the weights of
component classifiers on the basis of perplexity adaptively to each
test example.  This should be much more emphasized, while all the
component models are not genuinely novel.

Some descriptions are not precise enough.

- As I acknowledge above, the perplexity-based automatic weighting is
the key of the proposed method.  However, the formulation of the
weight lacks some information (see the questions below).

- The presentation of the results has a room for improvement.  Figures
3 and 4 are useless, because all the referenced information for the
discussion in Section 4.4 are also seen in Tables 1 and 2.  Using line
charts is also inappropriate because the results for different
datasets are inherently incomparable.

---

> ### Author Response · Authors · 2022-07-28
> **Thank you for your review and responses to your questions**
>
> Thank you for your review.
>
> We would like to give two justifications why we use auto-regressive models for the negative model and distribution model. However, we want to mention that we don't fully understand what the reviewer is saying here. So, to the best of our understanding what the reviewer is saying, the reason why we choose auto-regressive model for the negative model is that it enables training with only one class, while a nearest neighbor based model might not be easily trained with only one class. For the distribution model, we did ablations on other out-of-distribution metrics in addition to the perplexity score given by the auto-regressive model as shown in the subfigure (b) of figure 3 and figure 4 where nearest neighbor is similar to a simplified mahalanobis distance, and we empirically found that auto-regressive perplexity performs the best.
>
> Yes, two $P(s_2|s_1, Y)$ in eqn(4) equal respectively to the terms in eqn(3). We've corrected this in the revised version.
>
> When computing the perplexity, we concatenate the two sentences with a special separation token. By concatenating $s_1$ and $s_2$, the autoregressive model will first evaluate $P(s_1)$ and then evaluate $P(s_2|s_1)$ so they multiply to $P(s_1, s_2)$. However, if we iterate over $s_1$ and $s_2$ separately, it might only capture $P(s_1)$ and $P(s_2)$ without the dependence between them. We argue that the dependence between $s_1$ and $s_2$ is also an important sign of the distribution.
>
> Empirically we found that the distribution of perplexity exhibits a long-tail phenomenon or right-skewed. We hope that the skewed property of the Weibull distribution can help to capture this empirical observation.
>
> For M in equation (8), we clarify in our paper that we set M > 1000. Note that $\tau(\lambda (s_1, s_2))$ takes value of 0 or 1, so that when it takes 1 (meaning in-distribution), the GAP term will be negligible, otherwise when it takes 0, the GAP term will be the only term that remains while the first term vanishes no matter how large M is.
>
> Sorry, this is indeed a typo, should be $P() - \frac{1}{2}$, thank you for pointing that out.
>
> For the limitations, we do want to point out that paraphrase identification is an important topic in NLP, and there have been many top quality papers published on it, see [1],[2], [3], [4], [5]. We believe our work is important towards making progress in this field. While we understand that our proposed method cannot be immediately applied to other diverse NLP or CV tasks, our insights still apply as the need for using hard negatives do exist in many more applications. These include include Multiple Choices in Question Answering, 'Neutral' classes in NLI, hard negative mining in metric learning, etc. We do hope that our insight that manually designed hard negatives bring additional distribution bias can help to make more robust models in these other tasks.
>
> [1] Wuwei Lan, Siyu Qiu, Hua He, and Wei Xu. 2017. A Continuously Growing Dataset of Sentential Paraphrases. In Proceedings of the 2017 Conference on Empirical Methods in Natural Language Processing, pages 1224–1234, Copenhagen, Denmark. Association for Computational Linguistics.
>
> [2] Yinfei Yang, Yuan Zhang, Chris Tar, and Jason Baldridge. 2019. PAWS-X: A Cross-lingual Adversarial Dataset for Paraphrase Identification. In Proceedings of the 2019 Conference on Empirical Methods in Natural Language Processing and the 9th International Joint Conference on Natural Language Processing (EMNLP-IJCNLP), pages 3687–3692, Hong Kong, China. Association for Computational Linguistics.
>
> [3] Wenpeng Yin and Hinrich Schütze. 2015. Convolutional Neural Network for Paraphrase Identification. In Proceedings of the 2015 Conference of the North American Chapter of the Association for Computational Linguistics: Human Language Technologies, pages 901–911, Denver, Colorado. Association for Computational Linguistics.
>
> [4] Zhiguo Wang, Haitao Mi, and Abraham Ittycheriah. 2016. Sentence Similarity Learning by Lexical Decomposition and Composition. In Proceedings of COLING 2016, the 26th International Conference on Computational Linguistics: Technical Papers, pages 1340–1349, Osaka, Japan. The COLING 2016 Organizing Committee.
>
> [5] Gaurav Singh Tomar, Thyago Duque, Oscar Täckström, Jakob Uszkoreit, and Dipanjan Das. 2017. Neural Paraphrase Identification of Questions with Noisy Pretraining. In Proceedings of the First Workshop on Subword and Character Level Models in NLP, pages 142–147, Copenhagen, Denmark. Association for Computational Linguistics.

---

> > ### Comment · Reviewer_emd5 · 2022-08-08
> > **Thank you for your response. Let me clarify my question.**
> >
> > The authors answered that they chose auto-regressive model for the negative model because it enables training with only one class.  However, I'm not still convinced of this choice, and sorry that my question in the initial review was not precise enough.
> >
> > Let me explain my question more precisely.  An negative example is judged negative referring to the entire sentence, but not all the tokens inside are necessarily negative.  Suppose an negative example with $N$ tokens and only $k$-th token is troublesome which makes the entire sentence negative.  Among training signals (for auto-regressive models) drawn from this example, the first $(k-1)$ signals cannot be considered as negative because there is no negative factor and difference between positive examples, while $k$-th signal must be negative, and following signals are also affected by this troublesome token.  In other words, holistic treatment of each negative example, such as nearest neighbor or other sentence-level scoring, has a rationale, but analytic treatment, e.g., token-wise scoring, such as implemented by the proposed auto-regressive model, lacks the rationale.  I would like to know this before empirically testing it through an experiment.

---

> > > ### Author Response · Authors · 2022-08-08
> > > **Response to reviewer emd5**
> > >
> > > Thank you for your clarifications and sorry that we fail to accurately understand your concern previously. We now understand that, based on your clarification, sentence-level scoring can better capture cases where, say, only $k$th token makes the entire sentence pair negative, so that it might be more expressive than token-level scoring. We are happy to elaborate more on our choice of autoregressive method, both conceptually and empirically.
> > >
> > > Conceptually, since our methodology is aimed at separating the dependence on positive and negative samples for better generalization, we would like to reduce any implicit bias from contrasting positive and negative samples since the negative samples might be biased. We would like to caution that "only the kth token makes the sentence pair negative" might have to be learned from contrasting paraphrase pairs and non-paraphrase pairs. For the same example you gave, if we consider that the distribution of negative samples could be similar but non-paraphrase sentence pairs, the first k-1 tokens might turn out to be an in-class indicators instead since there is no negative factor in those tokens which makes the sentences similar. That's because without contrasting, it would be hard to pinpoint the part that makes a sample belong to the same class, so we use a reconstruction-like method (autoregressive) to keep all the information from the samples. We're not aware of any other objective that we can use with only one class of samples to tell if a tested sample is from that class.
> > >
> > > Empirically, discriminative training with only one class of samples can easily lead to a so-called "degenerate" solution where all the samples collapse to a constant in the sentence-level embedding space (so that in-class distance is minimized). On the other side, there is no such easily conceivable "degenerate" solution for the autoregressive method since all the tokens are given as non-trivial targets. If we force the model to operate in the sentence level instead of the token level, we're not sure how to provide a non-trivial target for the model to optimize for.

---

### Official Review · Reviewer_AaSh · 2022-07-11

**Rating:** 7
**Confidence:** 4
**Soundness:** 3 good
**Presentation:** 4 excellent
**Contribution:** 3 good

**Summary:**

This paper addresses the problem of paraphrase identification. The improvement is motivated by an observation where distributions of negative examples exhibits serious corpus-specific bias and do not generalize well across different corpora. To solve this problem, the paper starts with a Bayesian formulation of the paraphrase identification problem (formula (1)-(3)), and proposed three different ways to remedy the corpus-level bias, including out-of-distribution predictor (OODP), automatic ensemble of the (generative) positive and negative model (GAP), and an extra ensemble with a discriminative model (GAPX).

The experiments cover several different corpus and distribution shift scenarios (measured by RCA). Overall, results show an significant macro F1/ACC improvement of OODP/GAP/GAPX model over the IDP model, as well as BERT/RoBERTa-based baselines, which shows the benefit of the paper's proposed anti-biasing solutions.

**Questions:**

- Please comment on weakness point 2. I could have misunderstood something.
- It is not entirely clear to me why the distribution model (used to evaluated formula (5)) should be trained on the validation data -- isn't this trying to detect distribution shift from the training data?

**Limitations:**

Apart from the connection with Moore-Lewis filtering that the authors did not bring up, I also think the method could be further generalized and validated under cross-lingual scenarios -- for example, if $(s_1, s_2)$ are multilingual pairs and BART is substituted with mBART, the authors could use parallel data filtering task (https://www.statmt.org/wmt20/parallel-corpus-filtering.html) to further validate their findings.

**Strengths And Weaknesses:**

Strengths:

- The solution is very well-motivated by the authors' observation.
- The paper is overall well-written -- most parts of the proposed solutions are presented in a clear and intuitive way.
- The comparison is thorough, and shows clear improvement from the model across the board in out-of-distribution scenarios.

Weaknesses:

- The Bayesian formulation, while nicely laid out, is not a novel one. Specifically, Eqn. (3) has significant overlap with Moore-Lewis filtering (https://aclanthology.org/P10-2041.pdf). This connection is not pointed out in the paper.
- While theoretically-interesting, GAP/GAPX models do not show a lot of empirical improvement compared to the very naïve (even ill-formed, because it completely dropped the $P(w_2^{(i)}\mid s_1, Y=y, w_2^{(1:i-1)})$ term in the Bayesian formulation shown in (1)) OODP model, which is not clear to me why. Also, both GAP/GAPX models seem to have more components than OODP, so it's not clear whether GAP/GAPX is really worth it.

Some minor comments/suggestions:
- The paragraph of L34-51 is very long and verbose (especially since you are talking about a lot of modeling details without a formula). I would try to break it down to reflect the separation of different components (IDP -> OODP -> GAP/GAPX), as well as focusing more on providing a conciser summary of high-level intuitions.
- L101: as follow -> as follows
- L139-146: It would be clearer to have a separate subsection for IDP and OODP
- L203: You are referring to Eqn. (4) instead of (3.2).
- L242: "cross entropy"  -> what is this referring to? Eqn. (5)?

---

> ### Author Response · Authors · 2022-07-28
> **Thank you for your review and response to your questions**
>
> Thank you for your review.
>
> Thank you for pointing out the Moore-Lewis filtering method. Indeed our Bayesian formulation shares some similarity with that paper, but their method is used for selecting an in-domain corpus to train a language model while our formulation in Eq. 3 presents the relationship between the positive and negative model that allows for weighing positive and negative samples at inference time. We have made the connection to this work in our revised version.
>
> Regarding the advantage of GAP/GAPX over OODP, we would like to clarify that although OODP generalizes well to out-of-distribution scenarios, its in-distribution performance is far from satisfactory. As shown in Table 2, the average in-distribution performance of OODP is 60.4/70.4 (this might significantly hurt in-distribution applications) while GAPX can improve that to 78.1/78.5. Therefore, the advantage of GAPX is that it can automatically adapt to in-distribution and out-of-distribution scenarios by adjusting the weight it put on the negative samples.
>
> As to why we trained the distribution model with validation data, we really appreciate the reviewer's diligence here and realized we had made an editing mistake here, where Ln 155 and 156 - "Specifically, we hold back a set of validation data, comprising both positive and negative pairs, from D^s" - were meant for fitting the Weibull while the distribution model should be correctly described as being trained on the training data.
>
> We will also update our revised version with the reviewer's suggested edits. Ln 242's "cross entropy" refers to Eq. 3. We also appreciate the reviewer's comments on cross-lingual scenarios -- we will think about this but felt that this currently falls outside the task of paraphrase identification that we are tackling in this work.
>
> Again, we really appreciate the reviewer's diligence in reading our paper.

---

> > ### Comment · Reviewer_AaSh · 2022-08-08
> > **Thanks for the response!**
> >
> > Thanks for the authors' response.
> >
> > The clarification about OODP and GAPX is helpful for the readers to navigate the results. I now realize that you've talked about this during Section 4.5, but again within that paragraph you are jumping back and forth between several different points and it's a bit hard to follow. I'd suggest revision/rewrite of that paragraph to make the comparison between different models clearer. Those are your main results, after all :)
> >
> > But otherwise, after reading the response and other reviews, I think this should be accepted. I'm upgrading my score to 7.

---

### Official Review · Reviewer_mmQb · 2022-07-11

**Rating:** 5
**Confidence:** 4
**Soundness:** 3 good
**Presentation:** 3 good
**Contribution:** 3 good

**Summary:**

This paper explores the distribution shift problem in paraphrase identification.  The authors first verify that the distribution shift problem is mainly caused by the bias of negative examples.  To address this problem, they train two separate models, a positive model and a negative model, and combine them together during inference where the weights are dynamically decided by the distribution similarity between inference pair and training pairs. Experiments show that the proposed approach achieves good transfer learning ability.

**Questions:**

n/a

**Limitations:**

More comparison on datasets with in-pair data to show the generalization results over diverse tasks.

**Strengths And Weaknesses:**

Strengths:

The motivation is interesting. The proposed method is motivated by their findings that negative examples do not generalize well to out-of-distribution data. To address this model, they propose to separate positive examples and negative examples and train two models. The combination weights is adjusted during inference.


The proposed model outperforms baselines with a large margin.  Table 1 shows significant performance improvements over baselines.

Weaknesses:

Strong baselines are missing. Multi-task learning is also an important solution to address the distribution shift problem. It would be better to add a baseline that combines all tasks together. On the other hand, distribution shift is a traditional and important problem. The comparisons with related literature is required to show the effectiveness of the proposed method.

The proposed method is specifically designed for paraphrase identification with in-pair data. It is unclear whether the proposed method can affect diverse NLP tasks.

---

> ### Author Response · Authors · 2022-07-27
> **Thank you for your review, response to your questions and ask for clarifications**
>
> Thank you for your review.
>
> ### Multi-task learning as potential strong baselines:
> Would you mind clarifying your idea and if possible link us to papers on this? We ran some experiments on multi-task and did not see any gains (multi-tasking on paraphrase identification and natural language inference). However, we are not very sure how multi-tasking learning can apply here so it will be good to send us papers to clarify your comment. Here are the results that we have:
>
> Out of distribution (F1/Acc)
> |  | QQP -> PIT | PIT->QQP | QQP->WMT | PIT->WMT | PIT -> PAWS | PAWS->QQP | PAWS->PIT | PAWS->WMT | QQP->PAWS |
> | --- | ----------- |----------- |----------- |----------- |----------- |----------- |----------- |----------- |----------- |
> | BERT | 68.0/68.3 | 69.0/69.4 | 67.4/67.7 | 50.0/57.7 | 31.2/45.5 | 63.8/62.8 | 52.6/56.4 | 68.4/57.0 | 47.1/50.5 |
> | BERT (multitask with NLI) | 58.4/62.0 | 69.8/70.1 | 66.1/66.1 | 55.9/60.4 | 31.7/45.5 | 63.8/63.9 | 47.2/53.7 | 70.4/70.5 | 48.9/49.3 |
>
>
> ### Traditional distribution shift methods:
> We have looked into distribution shift and debiasing methods in NLP in Related Works Section 2.1. Among them, we benchmarked one of the most successful and representative method known as Expert Product (https://arxiv.org/abs/1909.03683) as a potential baseline. Other methods that we have found either failed to achieve any substantial improvement or share a similar idea with Expert Product. However, we did not observe any noticeable benefits of the Expert Product method for the task of paraphrase identification. It'll be great if you can point us to any other potential strong baselines that we have missed and we can include in our paper.
>
> ### Applications to diverse NLP tasks:
> We do want to point out that paraphrase identification is an important topic in NLP, and there have been many top quality papers published on it, see [1],[2], [3], [4], [5]. We believe our work is important towards making progress in this field.
>
> That said, our work is also relevant to other NLP tasks that involve the use of negative samples. For example, in Multiple Choice of Question Answering, researchers need to design confusing choices to complement the right answer. In inference tasks, researchers often need to design confusing samples with 'neutral' relation. Our finding is also relevant to the visual domain. For example, in metric learning, hard negative samples are utilized to encourage the model to learn a good metrics that can distinguish confusingly similar images. Although future work is needed to establish how our work may be applied to other tasks, we hope our findings that poorly designed negatives can introduce distribution bias will be useful.
>
> We'd like to address any further concerns that you may have.
>
> [1] Wuwei Lan, Siyu Qiu, Hua He, and Wei Xu. 2017. A Continuously Growing Dataset of Sentential Paraphrases. In Proceedings of the 2017 Conference on Empirical Methods in Natural Language Processing, pages 1224–1234, Copenhagen, Denmark. Association for Computational Linguistics.
>
> [2] Yinfei Yang, Yuan Zhang, Chris Tar, and Jason Baldridge. 2019. PAWS-X: A Cross-lingual Adversarial Dataset for Paraphrase Identification. In Proceedings of the 2019 Conference on Empirical Methods in Natural Language Processing and the 9th International Joint Conference on Natural Language Processing (EMNLP-IJCNLP), pages 3687–3692, Hong Kong, China. Association for Computational Linguistics.
>
> [3] Wenpeng Yin and Hinrich Schütze. 2015. Convolutional Neural Network for Paraphrase Identification. In Proceedings of the 2015 Conference of the North American Chapter of the Association for Computational Linguistics: Human Language Technologies, pages 901–911, Denver, Colorado. Association for Computational Linguistics.
>
> [4] Zhiguo Wang, Haitao Mi, and Abraham Ittycheriah. 2016. Sentence Similarity Learning by Lexical Decomposition and Composition. In Proceedings of COLING 2016, the 26th International Conference on Computational Linguistics: Technical Papers, pages 1340–1349, Osaka, Japan. The COLING 2016 Organizing Committee.
>
> [5] Gaurav Singh Tomar, Thyago Duque, Oscar Täckström, Jakob Uszkoreit, and Dipanjan Das. 2017. Neural Paraphrase Identification of Questions with Noisy Pretraining. In Proceedings of the First Workshop on Subword and Character Level Models in NLP, pages 142–147, Copenhagen, Denmark. Association for Computational Linguistics.

---

> > ### Comment · Reviewer_mmQb · 2022-08-06
> > **Thanks for your feedback**
> >
> > I wonder what the performance is if we directly merge all paraphrase data together.  Data augmentation is a widely-used choice to handle distribution shift. Distribution shift is a common problem for low-resource datasets. With the increasing number of training data, the distribution shift problem will gradually ease.
> >
> > Compared with the baseline starting from a model trained on raw texts, a baseline starting from a model trained on the combination of  existing paraphrase data will be more convincing.

---

> > > ### Author Response · Authors · 2022-08-08
> > > **Response to Reviewer mmQb**
> > >
> > > Thank you for your clarifications. Our work focuses on identifying the possible sources of biases that make models generalize poorly to out-of-distribution scenarios and coming up with solutions to overcome those possible sources of bias. We understand that out-of-distribution performances can be improved with other techniques like data augmentation to increase the amount of training data. However, adding more data can only improve the robustness of the model in a "gradual" way and obtaining high-quality data can be expensive. Furthermore, blindly adding data may retain, or even strengthen, the bias that causes the model to generalize poorly. As we can see from the preliminary experimental results below (which you asked for as well), merging datasets on the same scale does not always lead to a better results, and in the cases where it does, the improvement is not as significant as that provided by our proposed methodology.
> > >
> > > Out of distribution (F1/Acc)
> > > |  | QQP+PIT+PAWS -> WMT | QQP+PIT>WMT | PIT->WMT | PAWS->WMT | QQP->WMT |
> > > | --- | ----------- |----------- |----------- |----------- |----------- |
> > > | BERT | 70.3/70.3 | 65.1/65.1 | 50.0/57.7 | 68.4/57.0 | 67.4/67.7 |
> > > | GAPX| / | /| 74.4/74.5 | 76.4/76.4| 75.5/75.5 |
> > >
> > > |  | QQP+PIT->PAWS | QQP->PAWS | PIT->PAWS |
> > > | --- | ----------- |----------- |----------- |
> > > | BERT | 45.1/47.5| 47.1/50.5 | 31.2/45.5 |
> > > | GAPX| / | 52.3/54.3| 55.1/55.5 |
> > >
> > > We do want to conclude by saying that our work here is also not necessarily mutually exclusive with large scale pre-training regime. One of the most broadly used large scale pre-trained model known as CLIP (https://arxiv.org/abs/2103.00020) uses contrastive learning with positive and negative sample pairs. While extending our work here to CLIP falls outside the scope of this work, we hope that our research insight here that negative pairs produce bias can be considered in CLIP as well.
> > >
> > > We hope our new results and explanation address your concerns. Thanks again.

---

### Meta-Review · Area_Chair_Vcjx · 2022-08-23

**Recommendation:** Accept
**Confidence:** Less certain

**Metareview:**

This paper tackles a discriminative problem by a generative model, where the generation probabilities can be twisted to adjust negative samples’ weights.

Reviewers generally found the paper interesting. However, one concern is that the paper only considers the paraphrase-identification problem, which sounds narrow. It is expected that the approach may be generalized to different tasks.


**Award:**

No

---

### Decision · Program_Chairs · 2022-09-14

Accept